# High Immunogenicity to Influenza Vaccination in Crohn’s Disease Patients Treated with Ustekinumab

**DOI:** 10.3390/vaccines8030455

**Published:** 2020-08-14

**Authors:** Laura Doornekamp, Rogier L. Goetgebuer, Katharina S. Schmitz, Marco Goeijenbier, C. Janneke van der Woude, Ron Fouchier, Eric C.M. van Gorp, Annemarie C. de Vries

**Affiliations:** 1Department of Viroscience, Erasmus MC, University Medical Center Rotterdam, Postbus 2040, 3000 CA Rotterdam, The Netherlands; l.doornekamp@erasmusmc.nl (L.D.); k.schmitz@erasmusmc.nl (K.S.S.); m.goeijenbier@erasmusmc.nl (M.G.); r.fouchier@erasmusmc.nl (R.F.); 2Vaccination and Travel Clinic, Erasmus MC, University Medical Center Rotterdam, 3000 CA Rotterdam, The Netherlands; 3Department of Gastroenterology and Hepatology, Erasmus MC, University Medical Center Rotterdam, 3000 CA Rotterdam, The Netherlands; r.goetgebuer@erasmusmc.nl (R.L.G.); c.vanderwoude@erasmusmc.nl (C.J.v.d.W.); a.c.devries@erasmusmc.nl (A.C.d.V.); 4Department of Internal Medicine, Erasmus MC, University Medical Center Rotterdam, 3000 CA Rotterdam, The Netherlands

**Keywords:** ustekinumab, influenza vaccine, Crohn’s disease

## Abstract

Influenza vaccination can be less effective in patients treated with immunosuppressive therapy. However, little is known about the effects of ustekinumab; an anti-IL-12/23 agent used to treat Crohn’s disease (CD), on vaccination response. In this prospective study, we assessed immune responses to seasonal influenza vaccination in CD patients treated with ustekinumab compared to CD patients treated with anti-TNFα therapy (adalimumab) and healthy controls. Humoral responses were assessed with hemagglutinin inhibition (HI) assays. Influenza-specific total CD3^+^, CD3^+^CD4^+^, and CD3^+^CD8^+^ T-cell responses were measured with flow cytometry. Fifteen patients treated with ustekinumab; 12 with adalimumab and 20 healthy controls were vaccinated for seasonal influenza in September 2018. Seroprotection rates against all vaccine strains in the ustekinumab group were high and comparable to healthy controls. Seroconversion rates were comparable, and for A/H3N2 highest in the ustekinumab group. HI titers were significantly higher in the ustekinumab group and healthy controls than in the adalimumab group for the B/Victoria strain. Post-vaccination T-cell responses in the ustekinumab group were similar to healthy controls. One-month post-vaccination proliferation of CD3^+^CD8^+^ T-cells was highest in the ustekinumab group. In conclusion, ustekinumab does not impair immune responses to inactivated influenza vaccination. Therefore, CD patients treated with ustekinumab can be effectively vaccinated for seasonal influenza.

## 1. Introduction

Patients with inflammatory bowel disease (IBD) are frequently treated with immunomodulatory or immunosuppressive medication. Due to these therapies and the underlying inflammatory disease, they are at risk of more severe complications of infectious diseases [1]. Influenza causes significant morbidity and mortality in the general population [2] and the incidence of severe influenza is even higher in IBD patients, as demonstrated by higher rates of hospitalization (5.4% in IBD patients vs. 1.9% in healthy controls) [3]. Vaccination against influenza reduces the risk of infection in immunocompromised patients [4]. However, influenza vaccination may be less effective in patients treated with immunosuppressive therapies [5,6,7] and immunological mechanisms of the impaired vaccination response in IBD patients are often poorly understood [8].

Over the past decades, immunomodulatory and biologic therapies for the treatment of Crohn’s disease (CD) and ulcerative colitis (UC) have become widely available. Adalimumab is a frequently prescribed anti-TNFα agent that is administered subcutaneously and has proven efficacy for CD since 2006 [9]. The use of anti-TNFα agents and immunomodulators, especially when used combined, is associated with a lower serological response to influenza vaccination in both children and adults with IBD [5,6,7,10,11,12,13]. This is explained by the involvement of TNFα in B-cell and T-cell interactions to achieve adequate antibody production [14,15]. Ustekinumab, a human monoclonal antibody directed against the p40 subunit of interleukin (IL)-12 and IL-23 that normally binds to the interleukin-12 receptor β1 (IL-12Rβ1) of Th1 and Th17 cells, has more recently been approved as a treatment option for moderate-to-severe CD [9] and UC [16]. Although ustekinumab is effective and the safety profile reassuring [17,18], infections remain feared complications and preventive measures including annual influenza vaccination is currently advised by the European Crohn’s and Colitis Organisation (ECCO) guidelines [19]. Yet, little is known about the effects of ustekinumab on the immune responses to vaccinations.

Ustekinumab selectively inhibits IL-12 and IL-23 and thereby mainly Th1 and Th17 cell development [20]. However, IL-12Rβ1–mediated signaling via STAT3 and probably also STAT4, affected by ustekinumab treatment, plays a role in the generation of T follicular helper (T_FH_) cells [21]. As T_FH_ cells are important for the B-T cell interaction to generate high-affinity antibodies, humoral responses may be compromised [22]. In this study, we aim to investigate the humoral and cellular immune response after the inactivated 2018–2019 trivalent influenza vaccination (TIV) in adults with CD treated with ustekinumab (UST) compared to those treated with adalimumab (ADA) and healthy controls (HC).

## 2. Materials and Methods

### 2.1. Study Design and Population

We performed a prospective study on a selected cohort from a vaccination biobank in the Erasmus Medical Centre. All adult CD patients treated with either ustekinumab or adalimumab who wished to receive the seasonal influenza vaccination in September 2018 were asked to participate in the biobank study and were included following written informed consent. Healthcare workers who were offered the influenza vaccination for their occupation were selected from the biobank after age and sex matching to the CD patients and included as healthy controls.

### 2.2. Data Collection and Analysis

At baseline, informed consent forms were signed and medical history was collected from participants and electronic patient files. Medical IBD history was classified using the Montreal classification [23]. We collected medication use including dose at the moment of vaccination. Ustekinumab was routinely injected in a dose of 90 mg every eight weeks or 12 weeks and adalimumab in a dose of 40 mg once every two weeks, defined as a standard dose. More frequent injections were classified as escalated dose. Blood sampling was performed prior to the administration of the TIV. The 2018/2019 inactivated TIV (Influvac; Abbott Laboratories, Lake Bluff, IL, USA) contained 15 micrograms of HA antigen of each of the following influenza virus strains: A/Michigan/45/2015 (H1N1)pdm09-like virus; A/Singapore/INFIMH-16-0019/2016 (H3N2)-like virus; B/Colorado/06/2017-like virus (B/Victoria/2/87 lineage) and was administered intramuscularly in the deltoid. Patients were followed-up at one (T1), three (T3), and nine months (T9) post-vaccination. During each patient visit blood samples were collected in a BD Vacutainer^®^ Serum Separating Tubes II Advance Tubes and a BD Vacutainer^®^ CPT™ Cell Preparation Tube with Sodium Heparin^N^ (Thermo Fisher Scientific Inc., Waltham, MA, USA). Within 24 h after collection, serum samples were centrifuged and stored at –20 °C until further use. Peripheral blood mononuclear cells (PBMCs) were isolated by density gradient Ficoll separation and thereafter washed with phosphate buffer saline (PBS). Subsequently, PBMCs were counted and frozen in mononuclear cell medium with 10% dimethyl sulfoxide (DMSO) at a minimum of 2 × 10^6^ mononuclear cells per ampule. These samples were stored overnight in Nalgene^®^ Mr. Frosty™ Freezing Containers (Thermo Fisher Scientific Inc., Waltham, MA, USA) at −80 °C and transferred to liquid nitrogen thereafter.

### 2.3. Laboratory Assessments

#### 2.3.1. Hemagglutination Inhibition Assay

To assess antibody responses against the influenza virus vaccine strains, a hemagglutinin inhibition (HI) assay was performed simultaneously on all available serum samples, using a standard protocol [24,25]. Briefly, sera were pre-treated with neuraminidase from *Vibrio cholerae* (dilution of 1:5 of an in-house produced cholera filtrate), by incubation overnight at 37 °C and heat-inactivation for one hour at 56 °C. Nonspecific agglutination in sera was eliminated, if present, by incubating 15 parts of the serum-cholera filtrate mixture with one part 100% turkey erythrocytes for one hour at 4 °C. Due to the pre-treatment steps, a starting serum dilution of 1:10 was used for all experiments. Three hemagglutinin antigens, each representing a strain of virus contained in the vaccine, were added and twofold serial dilutions were made up to 1:20,480. The highest dilution of antiserum that was still able to block agglutination between test influenza viruses and 1% turkey erythrocytes was considered the HI titer.

#### 2.3.2. T-cell Proliferation Assay Using Flow Cytometry

Six doses of 2018/2019 inactivated TIV vaccine were dialysed (3 mL) with a slide-a-lyzer (Thermo Fisher Scientific Inc., Waltham, MA, USA) for contaminant removal to avoid interference in the T-cell proliferation assay. The amount of purified membrane glycoprotein subunit was analyzed with a bicinchoninic acid (BCA) assay (Thermo Fisher Scientific Inc., Waltham, MA, USA) and compared to undialysed vaccine content. If there was no difference in the amount of protein between dialysed and undialysed vaccine, we assumed no membrane protein was lost.

PBMCs were thawed at 37 °C and washed twice with IMDM (Thermo Fisher Scientific Inc., Waltham, MA, USA), supplemented with 2 mM L-glutamine, 100 U/mL penicillin (Lonza BioWhittaker^TM^, Basel, Switzerland) and 100 μg/mL streptomycin (Lonza BioWhittaker^TM^, Basel, Switzerland) (PSG) and 10% heat-inactivated fetal bovine serum (HI-FBS; Sigma-Aldrich, Saint Louis, MO, USA), further referred to as I10F. Subsequently, PBMCs were incubated with 50 U/mL Benzonase (Merck Millipore, Burlington, MA, USA) in I10F for 30 min at 37 °C, washed once and cultured overnight at a density of 1–3 × 10^5^ cells/well in RPMI-1640 supplemented with HI-FBS and PSG, further referred to as R10F. The next day, cells were washed once with PBS and labeled with 600 nM CFSE (in PBS) for 5 min at 37 °C. Afterward, PBMCs were washed with R10F, plated at a density of approximately 1.5 × 10^5^ cells per well in R10F, and cultured for five days. Per donor and time point three wells were left unstimulated, while three wells were stimulated with 100 ng/well of the dialysed purified membrane glycoprotein subunit preparations of the 2018/2019 TIV [26]. Concanavalin A (ConA) was used as a positive control at a concentration of 5 µg/mL. Five days after stimulation PBMCs were stained for CD3, CD4, and CD8. Briefly, cells were washed once with PBS containing 2 mM EDTA and 0.05% BSA (FACS buffer) and then stained for 15 min at 4 °C in FACS buffer with the following monoclonal antibody-fluorochrome conjugates: CD3/APC Cy7 (1:50 dilution, BD Pharmingen), CD4/V450 (1:50 dilution, BD Horizon), and CD8/PE-Cy7 (1:25 dilution, eBioscience). After staining, cells were washed twice with FACS buffer and flow cytometry was performed with a BD FACSLyric^TM^ flow cytometer (BD Bioscience, Franklin Lakes, NJ, USA).

### 2.4. Outcomes and Parameters

Functional antibody responses were assessed with the HI assay. The assay was performed in duplo, and geometric mean titers were calculated. For calculation purposes, HI titers <10 were adjusted to 1. From these results, the following outcomes were calculated: (1) seroprotection rate: the percentage of participants per study group with an antibody titer above 40, which is considered the best surrogate correlate of protection [27]; (2) seroconversion rate: the percentage of participants in the study group that had at least a fourfold increase of the post-vaccination antibody concentration when compared to the pre-vaccination antibody concentration; (3) geometric mean titers (GMT) per time point per study group. We corrected for high pre-vaccination antibody titers, using a log10 transformation of GMTs and a linear regression formula described by Beyer and colleagues [28], which results in a “reset” of pre-vaccination antibody titers to zero. Data were back log-transformed to show interpretable results.

Cellular responses were assessed by the proliferation of influenza-specific CD3^+^, CD4^+^, and CD8^+^ T-cells. Stimulation indexes (SI) were calculated by dividing the percentage of proliferated cells in stimulated samples by the percentage of proliferated cells in unstimulated samples per donor, time point, and T-cell subset (total CD3, CD4, or CD8).

### 2.5. Data Analysis

FACS data were analyzed with FlowJo version 10.6.1. Gating strategies used for analysis are shown in Appendix A. We set the mean background of proliferation in unstimulated samples to 1.5 and applied the same gating strategy to stimulated samples. SPSS version 24 was used for data analysis. A Shapiro-Wilk test was used to assess the normality of distributions. A comparison of parametric continuous variables between the three groups was done using one-way ANOVA and of non-parametric continuous variables using Kruskal–Wallis tests. A comparison of non-parametric continuous variables between two groups was done using Mann–Whitney U tests. Fisher’s exact tests were used for comparing differences in categorical variables. To prevent finding significances due to multiple testing, we only performed testing between two groups when the comparison between the three groups showed a *p*-value of < 0.10. A *p*-value < 0.05 was considered significant. Outliers were detected with the Tukey’s box-plot method which defines outliers as being outside the interquartile interval (Q1–1.5∙IQR, Q3 + 1.5∙IQR). Missing data were excluded per variable. GraphPad Prism version 5.0 for Windows (GraphPad Software, San Diego, CA, USA) was used to create figures. The ustekinumab group is depicted in the figures with a blue square, the adalimumab group with a green triangle and the healthy controls with a grey circle.

### 2.6. Ethical Considerations

All subjects gave their informed consent for inclusion before they participated in the study. This study has been exempted from medical ethical approval requirements by the Medical Ethical Research Committee of the Erasmus Medical Center on November 13, 2017, due to the biobank format of the Vaccination Cohort study (COVA study, MEC-2014-398). The study has been conducted according to the principles of the declaration of Helsinki (64th WMA General Assembly, Fortaleza, Brazil, October 2013).

## 3. Results

Forty-seven subjects were enrolled in this study between September 2018 and November 2018. We studied the 2018–2019 TIV vaccine response of 47 individuals in three different study groups: 15 CD patients using ustekinumab, 12 CD patients using adalimumab, and 20 healthy controls with influenza vaccination history. Demographic baseline characteristics were comparable between the three groups and described in Table 1. The average median age of the total study population was 39 years (IQR 29–50) and 57 percent was female. The median duration of the use of adalimumab was 32 months, and 19 months for ustekinumab (*p* = 0.022). In the ustekinumab group, one patient was injected every seven weeks and one patient every six weeks. In the adalimumab group, two patients were injected weekly, two patients every 10 days, and once every four weeks. Three patients in the ustekinumab group additionally used an immunomodulator (thiopurines or methotrexate) compared to two patients in the adalimumab group. Montreal classification, use of co-medication, and influenza vaccination history did not differ significantly between the three groups.

### 3.1. Humoral Immune Response

#### 3.1.1. Seroprotection Rates

Pre-vaccination seroprotection rates for all three strains were not significantly different between the groups in Table 2. Seroprotection rates to the H3N2 strain one-month post-vaccination were 100 percent in all three groups and remained 100 percent three months post-vaccination in healthy controls and the ustekinumab group. In the adalimumab group, seroprotection rates were lower three months post-vaccination compared to the other two groups, reaching borderline significance (81.8%, *p* = 0.056). Seroprotection rates to the H1N1 strain were higher than 90.0 percent one-month post-vaccination and at least 78.6 percent three months post-vaccination for the three study groups and did not differ significantly Table 2. Pre- and post-vaccination titers were lowest to the B/Victoria strain, especially in the adalimumab group T1 and T3: 63.6%, however, there was no significant difference between study groups.

#### 3.1.2. Seroconversion Rates

Seroconversion rates to the H3N2 strain were significantly different in the three groups at three months post-vaccination (T3: *p* = 0.014, Table 3) and borderline significant at one-month post-vaccination (T1: *p* = 0.064). The ustekinumab group had higher seroconversion rates compared to the adalimumab group (T3: *p* = 0.015, Table 3) and the healthy controls (T1: *p* = 0.038, T3: *p* = 0.035, Table 3). Seroconversion rates to the other influenza vaccine strains in the three study groups were highest in the ustekinumab group and lowest in the adalimumab group, although this reached no significance.

#### 3.1.3. Antibody Titers

The post-vaccination antibody titers in the ustekinumab group were comparable to those of the healthy controls. In the adalimumab group, geometric mean titers (GMT) were lower compared to the other two groups for all influenza vaccine strains at both T1 and T3, except for the H1N1 strain three months post-vaccination Table 4, Figure 1. This reached significance for the B/Victoria strain at both T1 and T3, when comparing the three groups (T1: *p* = 0.031 and T3: *p* = 0.028, Table 4, Figure 2) and specifically the ustekinumab and adalimumab group (*p* = 0.028 and *p* = 0.009, Table 4) respectively.

As pre-vaccination titers in the ustekinumab group were significantly lower than in the healthy controls and the adalimumab group (*p* = 0.013), we studied antibody titers after correction for high pre-vaccination titers in the latter two Table 4. Post-correction antibody titers at T3 for the H3N2 strain were significantly lower in the adalimumab group compared to healthy controls and the ustekinumab group (*p* = 0.041, Table 4). For the B/Victoria strain, post-correction antibody titers were significantly higher for both Table 1, Table 3 in the ustekinumab group compared to the other two groups (T1: *p* = 0.014, T3: *p* = 0.015, Table 4).

### 3.2. Cellular Immune Response

T-cell proliferation was studied per group, per time point and per T-cell subset (example shown in Appendix A). In general, stimulation indexes showed a pattern of increased proliferation from baseline to T1 and T3 (except CD3^+^CD8^+^ response in healthy controls) and a decrease between T3 and T9 (except the CD3^+^ and CD3^+^CD8^+^ response in the ustekinumab group) Figure 3. In all three groups, baseline CD3^+^ and CD3^+^CD4^+^ responses were low (mean SI < 1.36). However, CD3^+^CD8^+^ baseline responses were relatively high (mean SI >1.49). When comparing time points and T-cell subsets, no significant differences were found between the three study groups. However, when we compared the groups one by one, we found a significantly higher CD3^+^CD8^+^ response one month after vaccination for the ustekinumab group compared to healthy controls (*p* = 0.025).

Overall, 95% confidence intervals were large and a few donors from all groups showed exceptional high responses (Figure 3). When excluding these outliers, stimulation indexes were significantly different between the three groups one-month post-vaccination in the CD3^+^CD8^+^ subset (*p* = 0.031) in favor of the ustekinumab group (UST vs. HC, *p* = 0.009) Appendix A.

### 3.3. Correlation Between Humoral and Cellular Immune Response

To assess a possible relationship between the humoral and cellular immune responses, we calculated correlations between HI assay titers (GMTs for the three different vaccine strains) and the stimulation indexes (for the three different subsets of T-cell populations) Appendix A. The highest Spearman correlation coefficient was found between the GMTs for the H1N1 strain and the SI for the CD3^+^ T-cells (R = 0.278, *p* = 0.002).

## 4. Discussion

Influenza vaccination is recommended in IBD patients according to international guidelines; however, immunomodulatory or immunosuppressive treatment may impair vaccine responses. This prospective cohort study showed that B-cell as well as T-cell responses to inactivated TIV in patients with CD during ustekinumab treatment were maintained and not impaired compared to healthy controls. Patients treated with ustekinumab had comparable seroprotection rates post-vaccination as healthy controls and better-sustained seroprotection rates to the H3N2 strain than patients treated with adalimumab. Seroconversion rates were also higher in the ustekinumab group compared to healthy controls and the adalimumab group at three months post-vaccination for the H3N2 strain. After correction for high pre-vaccination titers using a linear regression formula described by Beyer et al. [28]. Post-correction, post-vaccination titers were significantly higher in the ustekinumab group compared to the adalimumab group and healthy controls for the B/Victoria strain. Cellular immune responses in the ustekinumab group were not impaired either. The CD8^+^ T cell response one-month post-vaccination was even significantly higher than in healthy controls.

To our knowledge, this is the first study that shows the immune response to vaccination in CD patients treated with ustekinumab. Our results are in line with a previous study in psoriasis patients treated with ustekinumab. This study showed no differences in the immune response to pneumococcal or tetanus toxoid vaccinations in patients treated with ustekinumab compared to controls [29]. In another study, higher antibody responses to hepatitis B virus vaccination were found in patients treated with ustekinumab compared to patients treated with infliximab or adalimumab [30]. Immune response to influenza vaccination in patients treated with ustekinumab have not been reported yet. Our results indicate that blocking IL-12 and IL-23 does not influence immune responses to vaccination as has been previously hypothesized [31]. T_FH_ cells could still be generated, as studies in IL-12Rβ1-deficient adults have shown that the level of T_FH_ cells was not reduced in the absence of IL-12Rβ1 [32]. Alternatively, if the generation of T_FH_ cells is impaired due to the effect of a lacking signal to IL-12Rβ1 on the STAT3 (and 4) pathway, extrafollicular T helper cells might take over T_FH_ cells functions [21,33].

Although measured against different influenza strains, the HI assay responses in our healthy controls were comparable to those in previous studies, or even higher [34,35]. Higher GMTs can be explained by the influenza vaccination history in our study population Seroconversion rates might be lower than in non-immune populations due to high pre-vaccination titers. Although antibody titers only increase slightly after repeated annual influenza vaccination, they still prevent laboratory proven influenza infections [35]. Several previous studies have shown decreased immune responses to influenza vaccination in IBD patients using anti-TNFα agents [6,7,11,12,13]. This is in line with our results from the HI assay, but not reflected by our T-cell proliferation data.

We found no previous studies on cellular responses after influenza vaccination in adult IBD patients. In children with IBD, it was shown, in line with our data, that lymphocyte proliferation in general and after stimulation with tetanus antigen and adenovirus antigen was not impaired by several immunosuppressive therapies [36]. For T-cell proliferation assays in liver transplant recipients who were vaccinated for seasonal influenza higher SI indexes in healthy controls and patients were reported compared to our data [26]. However, due to the use of a thymidine assay to measure the influenza-specific T-cell response at that time, the results might not be comparable to our flow cytometry results. A study investigating T-cell responses after influenza vaccination reported short-lived CD4^+^ T cell responses when PBMCs were stimulated with live (attenuated) virus strains [35]. This is in contrast with our data showing that the T-cell response was still high (or even highest) three months post-vaccination.

In this era of new therapeutic targets and personalized treatment, the immune response to vaccination might be an extra aspect influencing the choice of therapy, in addition to commonly weighed factors such as effectiveness, safety, and costs. Combination therapy with anti-TNFα agents and an immunomodulatory agent is more effective for the treatment of CD than monotherapy, most likely due to both suppression of immunogenicity and the additive effects of the two drugs to reach disease remission [37]. However, this combined strategy is also associated with a higher risk of infections [38] and may have a negative impact on immune responses to vaccination [5,11,12]. Several ways to improve the influenza vaccination response during anti-TNFα therapy have been investigated. A booster vaccination failed to show better protection rates [7,39] and timing relative to infliximab infusion neither showed to affect serological protection [13]. Recently, a study found that four times higher dose vaccination resulted in higher antibody responses to influenza vaccination compared to the standard dose, without leading to more adverse effects [40]. Yet, “high dose” vaccination is currently only recommended by American guidelines for patients aged 65 years or older [41]. Current evidence does not support the use of immunomodulatory agents combined with ustekinumab [17]. Similar to our results in the ustekinumab group, a recent study showed that immune responses to influenza vaccination in patients treated with vedolizumab, a monoclonal antibody against the α4β7 integrin, were not altered either [40]. Interestingly, the immune response to an enterally administered vaccine was impaired during treatment with vedolizumab, possibly reflecting the gut-selective action of this therapy [42].

The ECCO recommends routine influenza vaccination of patients on immunomodulators [19]. However, reported influenza vaccination uptake rates among IBD patients are low (28 to 61%) [1,43,44,45], amongst others due to concerns about effectiveness and their unawareness of the recommendation [1,43,44]. With our results, we provide evidence for the high immunogenicity of influenza vaccination in CD patients treated with ustekinumab. As vaccination check-ups and active vaccination recommendations by treating physicians or supportive nurses are associated with improved vaccination uptake [43,45], we strongly support involved nurses and physicians to recommend annual influenza vaccination to their patients treated with ustekinumab. This advice is similar for CD patients treated with adalimumab because even though anti-TNFα treatment is associated with a lower serological response, the CD4^+^ and CD8^+^ T-cell responses showed to be non-inferior in this study.

Few limitations need to be taken into account with the interpretation of our results. First, this study is hampered by a small sample size and the number of patients on combination therapy with an immunomodulatory agent was too small to do a subgroup analysis. Since the lowest influenza vaccine responses in IBD patients are reported in patients using combination therapy with an anti-TNFα agent and an immunomodulatory agent [5,11,12], this would have been an interesting addition. However, as immunomodulatory use was equally distributed amongst the patient groups, this will not have affected their comparison. The heterogeneity in the dose of medication in both adalimumab and ustekinumab users at the moment of vaccination might have influenced the results. Furthermore, in this study CD patients were treated with adalimumab. Although anti-TNFα agents are comparable in efficacy and side effects, vaccine responses may differ, and cannot be generalized to other anti-TNFα agents than adalimumab. There was a significantly higher baseline GMT for the A/H3N2 strain in healthcare workers compared to the ustekinumab group. As the influenza vaccination history was comparable between the three study groups, this might be explained by higher exposure to influenza in healthcare workers [46]. By using a linear regression formula, we were able to correct for this possible confounder. As the composition of influenza vaccinations change annually it is hard to compare our results one by one with previous and future studies. While we broadly examined influenza-specific immune responses by studying both humoral and cellular responses, in-depth details remain to be elucidated. The HI assays showed that functional antibodies are present in ustekinumab-treated patients, but we cannot conclude anything about the isotypes of the antibody response. With the T-cell proliferation, we showed comparable proliferation of influenza-specific CD4^+^ and CD8^+^ T cells in all study populations, but the effector functions of CD4^+^ and CD8^+^ T cells are still unknown. Therefore, the performance of an intracellular cytokine staining would have been of additional value. Lastly, we compared immunological outcome measures between groups and could therefore not directly conclude morbidity due to influenza infections in these cohorts.

## 5. Conclusions

In this study, we demonstrated that CD patients treated with ustekinumab have adequate B- and T-cell responses to influenza vaccination. Therefore, our data support the plea for influenza vaccination in CD patients treated with ustekinumab to protect them from severe infections.

## Figures and Tables

**Figure 1 vaccines-08-00455-f001:**
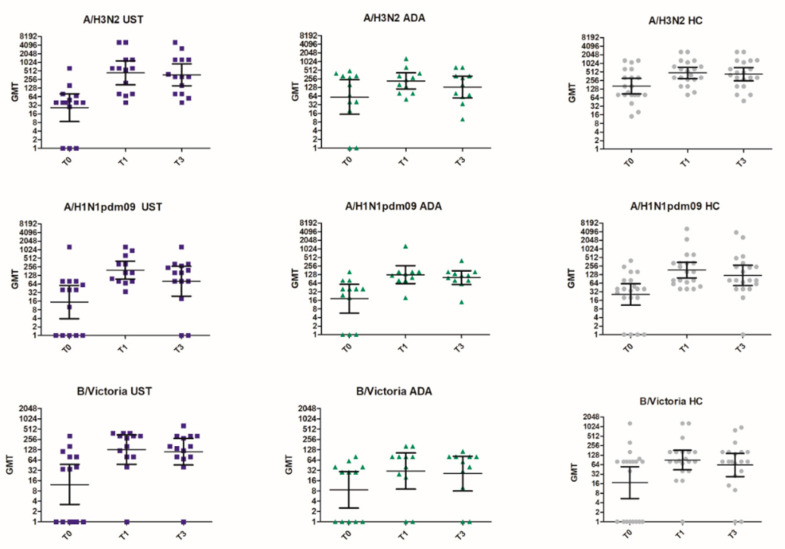
HI titers for each participant to influenza A/H3N2, A/H1N1pdm09, and B/Victoria vaccination per strain and study group. T0 = pre-vaccination, T1 = one-month post-vaccination, T3 = three months post-vaccination, T9 = nine months post-vaccination. UST = ustekinumab group (blue square), ADA = adalimumab group (green triangle), HC = healthy controls (grey circle). Geometric mean titers (GMT) and 95% confidence intervals are shown.

**Figure 2 vaccines-08-00455-f002:**
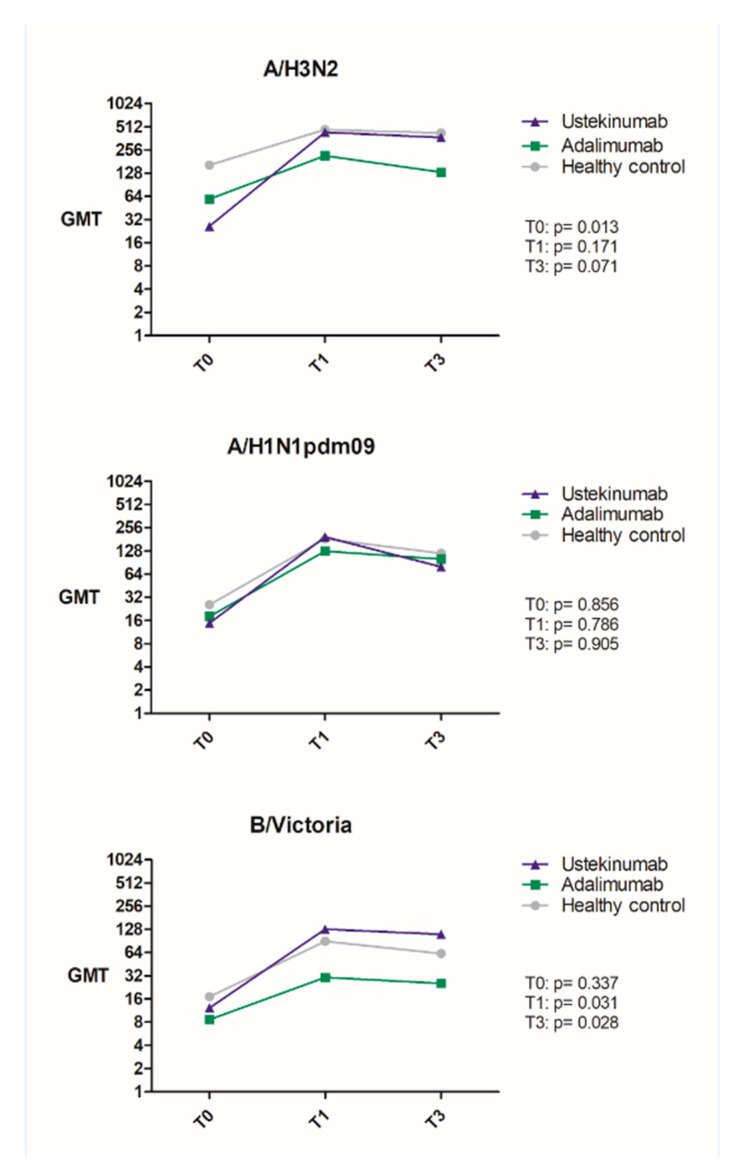
Dynamics of geometric mean HI titers (GMT) to influenza A/H3N2, A/H1N1pdm09, and B/Victoria vaccination according to study groups. T0 = pre-vaccination, T1 = one-month post-vaccination, T3 = three months post-vaccination. Comparisons between groups were tested using Kruskal–Wallis tests. A *p*-value < 0.05 indicates statistical significance.

**Figure 3 vaccines-08-00455-f003:**
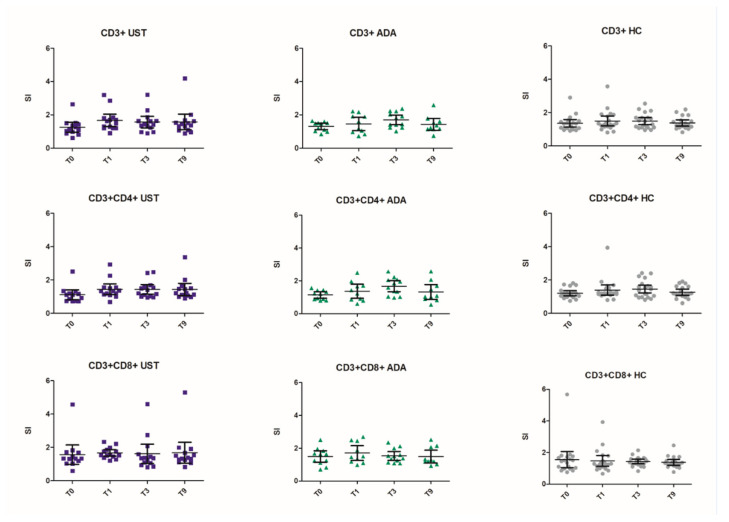
Stimulation indexes for each participant in the three study groups per T-cell subset. T0 = pre-vaccination, T1 = one-month post-vaccination, T3 = three months post-vaccination, T9 = nine months post-vaccination. SI = stimulation index. UST = ustekinumab group (blue square), ADA = adalimumab group (green triangle), HC = healthy controls (grey circle). CD3+ = CD3^+^ T-cells, CD4+ = CD4^+^ T-cells (T helper cells), CD8+ = CD8^+^ T-cells (cytotoxic T-cells). 95% confidence intervals are shown.

**Table 1 vaccines-08-00455-t001:** Baseline characteristics.

Baseline Characteristics	UST	ADA	HC	Difference Between Groups
	*n* = 15	*n* = 12	*n* = 20	Sig. (*p*-value)
**Gender**				
Female, *n* (%)	11 (73.3)	5 (41.7)	11 (55.0)	0.260 ^†^
Pregnant, *n* (%)	1 (9.1)	2 (40.0)	0 (0)	0.079 ^†^
**Age**				
Median, years (IQR)	36 (26–56)	45 (28–59)	36 (29–49)	0.688 ^‡^
**Country of birth**				
Netherlands, *n* (%)	13 (86.7)	11 (91.7)	19 (95.0)	0.808 ^†^
**BMI**				
Mean, kg/m2 (SD)	24.5 (4.6)	25.3 (5.2)	24.0 (4.3)	0.723 ^§^
**Lifestyle**				
Smoker, *n* (%)	5 (33.3)	1 (8.3)	3 (15.0)	0.201 ^†^
Alcohol, *n* (%)	9 (60.0)	7 (58.3)	19 (95.0)	**0.016 ^†^**
**Duration of CD**				
Median, years (IQR)	15 (9–25)	14 (8–35)	NA	0.845
**Disease Location**				
Teminal ileum (L1)	3 (20.0)	2 (16.7)		
Colon (L2)	1 (6.7)	1 (8.3)		
Ileocolon (L3)	8 (53.3)	7 (58.3)		
Ileocolon and upper GI (L3+L4)	3 (20.0)	2 (16.7)	NA	1.000 ^†^
**Disease Behavior**				
Nonstricturing, nonpenetrating (B1)	4 (26.7)	4 (36.4)		
Stricturing (B2)	7 (46.7)	6 (54.5)		
Penetrating (B3)	4 (26.7)	1 (9.1)	NA	0.666 ^†^
Perianal disease (*p*)	4 (26.7)	3 (25.0)	NA	1.000 ^†^
**Duration medication,**				
Median, months (IQR)	13 (5–19)	32 (15–82)	NA	**0.022**
**Dose medication**				
Standard dose	13 (86.7)	7 (66.7)		
Escalated dose	2 (13.3)	4 (33.3)	NA	**0.357 ^†^**
**Immunosuppressive**				
**co-medication * *n* (%)**				
None	9 (60.0)	7 (58.3)		
Low dose corticosteroids	2 (13.3)	3 (25.0)		
High dose corticosteroids	1 (6.7)	0 (0.0)		
Methotrexate	2 (13.3)	0 (0.0)		
Thiopurines	1 (6.7)	2 (16.7)	NA	0.643 ^†^
**Influenza vaccine history, *n* (%)**				
never before	3 (20.0)	3 (25.0)	6 (30.0)	
once before (2017)	0 (0.0)	2 (16.7)	2 (10.0)	
twice before (2016, 2017)	1 (6.7)	0 (0.0)	1 (5.0)	
thrice before (2015–2017)	0 (0.0)	1 (8.3)	1 (5.0)	
more than thrice before	5 (33.3)	5 (41.7)	4 (20.0)	
at least once, but not 2017	6 (40.0)	1 (8.3)	6 (30.0)	0.537 ^†^

Percentages within study groups. *T*-tests were used to calculate differences between continuous variables, chi-square tests were used for categorical variables. UST = ustekinumab group, ADA = adalimumab group, HC = healthy controls. CD = Crohn’s Disease, GI = gastrointestinal, NA = not applicable. * used while vaccinated or during the 3 months before. Low-dose corticosteroids = prednisone < 10 mg/day or budesonide (<9 mg/day). High-dose corticosteroids = prednisone ≥ 10 mg/day (at least 14 consecutive days or 700 mg total). ^†^ Fisher’s exact test, ^‡^ Kruskal–Wallis test, ^§^ one-way ANOVA, ^¶^ Mann–Whitney U test.

**Table 2 vaccines-08-00455-t002:** Seroprotection rates per study group (% HI–titers ≥ 1:40).

Influenza Strains		UST	ADA	HC	Overall	UST vs. HC	UST vs. ADA	ADA vs. HC
		*%*	*%*	*%*	*p-Value*	*p-Value*	*p-Value*	*p-Value*
**A/H3N2**	**T0**	71.4	75.0	90.0	0.328			
	**T1**	100	100	100	1			
	**T3**	100	81.8	100	0.056	1	0.183	0.118
**A/H1N1pdm09**	**T0**	57.1	58.3	55.0	0.982			
	**T1**	91.7	90.0	100	0.379			
	**T3**	78.6	90.9	90	0.561			
**B/Victoria**	**T0**	42.9	33.3	60.0	0.311			
	**T1**	92.3	63.6	85.0	0.170			
	**T3**	92.9	63.6	75.0	0.202			

UST = ustekinumab group, ADA = adalimumab group, HC = healthy controls. Significances were calculated with Fisher’s exact tests.

**Table 3 vaccines-08-00455-t003:** Seroconversion rates per study group (% ≥4-fold increase).

Influenza Strains		UST	ADA	HC	Overall	UST vs. HC	UST vs. ADA	ADA vs. HC
		*%*	*%*	*%*	*p-Value*	*p-Value*	*p-Value*	*p-Value*
**A/H3N2**	**T0–T1**	69.2	27.3	30.0	0.064	**0.038**	0.100	1
	**T0–T3**	71.4	18.2	30.0	**0.014**	**0.035**	**0.015**	0.676
**A/H1N1pdm09**	**T0–T1**	75.0	40.0	50.0	0.288			
	**T0–T3**	50.0	36.4	45.0	0.863			
**B/Victoria**	**T0–T1**	61.5	27.3	35.0	0.227			
	**T0–T3**	50.0	27.3	30.0	0.520			

UST = ustekinumab group, ADA = adalimumab group, HC = healthy controls. Significances were calculated with Fisher’s exact test.

**Table 4 vaccines-08-00455-t004:** Geometric mean antibody titers (GMT) per study group per time point.

Influenza Strains		UST	ADA	HC	Overall	UST vs. HC	UST vs. ADA	ADA vs. HC
A/H3N2	*GMT*				*p-Value*	*p-Value*	*p-Value*	*p-Value*
	T0	26	59	163	**0.013**	**0.008**	0.252	0.586
	T1	437	215	474	0.171			
	T3	372	132	427	0.071			
	***Post-correction GMT***							
	T1	203	75	141	0.159			
	T3	132	35	85	**0.041**	0.396	**0.025**	**0.036**
**A/H1N1pdm09**	***GMT***							
	T0	15	18	26	0.856			
	T1	195	127	184	0.786			
	T3	80	101	120	0.905			
	***Post-correction GMT***							
	T1	107	60	91	0.261			
	T3	27	29	33	0.947			
**B/Victoria**	***GMT***							
	T0	12	9	17	0.337			
	T1	129	30	90	**0.031**	0.073	**0.028**	0.306
	T3	111	26	62	**0.028**	0.125	**0.009**	0.220
	***Post-correction GMT***							
	T1	53	13	31	**0.014**	**0.043**	**0.005**	0.197
	T3	42	10	21	**0.015**	**0.036**	**0.006**	0.227

UST = ustekinumab group, ADA = adalimumab group, HC = healthy controls. GMT = geometric mean antibody titer. Post-correction GMT = transformed post-vaccination GMTs corrected for high pre-vaccination titers. Significance between GMT and post-correction GMT values was calculated with a Kruskal–Wallis test. If the Kruskal–Wallis test showed a significant difference, differences between separate groups were calculated with Mann–Whitney U-tests.

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
