# Peer review of "High Immunogenicity to Influenza Vaccination in Crohn’s Disease Patients Treated with Ustekinumab"

_vaccines, 2020, doi:10.3390/vaccines8030455_

Round 1
Reviewer 1 Report
The article reported the immunogenicity of Influenza vaccination in Crohn’s disease patients treated with Ustekinumab or adalimumab. Humoral and cellular immune responses were examined at the baseline, one month, three months and 9 months after vaccination. Of interest, not only ustekinumab did not impair immune responses to influenza vaccination, in some cases, ustekinumab treatment showed enhanced antibody response compared to the healthy group. In comparison, patients receiving adalimumab had reduced antibody responses. While the study is important and observation is interesting, there are issues need to be addressed before the article can be published.
- A specific concern regarding the study design is that the median duration of use of adalimumab was 32 months and 19 months for ustekinumab. The study has not examined the impact of prolonged used of ustekinumab in vaccination. Unlike the authors’ believe that IL-12/IL-23 do not influence Tfh responses, there are ample evidence suggest that IL-12 has important role in Tfh differentiation. IL-12 receptor β1 deficiency alters in vivo T follicular helper cell response in humans (Blood. 2013 Apr 25; 121(17): 3375–3385). This knowledge should be reflected in the introduction and discussion sections.
- Fig 1 shows that the baseline of H3N2 in ustekinumab group is much lower than the control group. What causes the difference? Due to the frequency of vaccination or this may reflect a reduced memory response in ustekinumab group?
- With the understanding that IL-12/IL-23 play a role in regulating Tfh responses, it is critical to examine the isotypes of antibody response in all treatment group. While the total HI may not be affected by ustekinumab treatment, the antibody composition in different treatment groups may be very different.
- The authors also examined cellular responses by examining cell proliferation (Fig 3), which provided very limited information. Compared to the baseline, even healthy control group did not show enhanced proliferation, which is concerning. Intracellular cytokine staining is a preferred method to examine effector functions of CD4 and CD8 T cell responses.
- All figures should include statistical analysis with proper labels.
Author Response
The article reported the immunogenicity of Influenza vaccination in Crohn’s disease patients treated with Ustekinumab or adalimumab. Humoral and cellular immune responses were examined at the baseline, one month, three months and 9 months after vaccination. Of interest, not only ustekinumab did not impair immune responses to influenza vaccination, in some cases, ustekinumab treatment showed enhanced antibody response compared to the healthy group. In comparison, patients receiving adalimumab had reduced antibody responses. While the study is important and observation is interesting, there are issues need to be addressed before the article can be published.
Response of the authors:
We would like to thank reviewer #1 for a thorough review of our manuscript and providing very valuable comments. We addressed them one-by-one hereafter.
A specific concern regarding the study design is that the median duration of use of adalimumab was 32 months and 19 months for ustekinumab. The study has not examined the impact of prolonged used of ustekinumab in vaccination. Unlike the authors’ believe that IL-12/IL-23 do not influence Tfh responses, there are ample evidence suggest that IL-12 has important role in Tfh differentiation. IL-12 receptor β1 deficiency alters in vivo T follicular helper cell response in humans (Blood. 2013 Apr 25; 121(17): 3375–3385). This knowledge should be reflected in the introduction and discussion sections.
Response of the authors:
We thank reviewer for the helpful literature provided and agree that this is important to include in our manuscript.
As ustekinumab was only recently approved for treatment of Crohn’s disease we were not able to draw any conclusions of the impact of prolonged use of ustekinumab. Since Adalimumab was introduced 10 years before ustekinumab, the median exposure duration was expected to be longer in the Adalimumab group. However, we expect the ustekinumab group with a median duration of 19 months to be sufficiently exposed to evaluate the effect of the drug on the vaccination response.
We now discuss the role of T follicular helper cells in the introduction and discussion section and added the suggested reference (now ref 21).
The final paragraph of the introduction has changed to: “Ustekinumab selectively inhibits IL-12 and IL-23 and thereby mainly Th1 and Th17 cell development [20].However, IL-12Rβ1–mediated signaling via STAT3 and probably also STAT4, affected by ustekinumab treatment, plays a role in the generation of T follicular helper (TFH) cells {Schmitt, 2013}. As TFH cells are important for the B-T cell interaction to generate high-affinity antibodies, humoral responses may be compromised.”
The discussion section (line 299-304) is supplemented with: “TFH cells could still be generated, as studies in IL-12Rβ1-deficient adults have shown that the level of TFH cells was not reduced in the absence of IL-12Rβ1 {Ma, 2012}. Alternatively, if the generation of TFH cells is impaired due to the effect of a lacking signal to IL-12Rβ1 on the STAT3 (and 4) pathway, extra follicular T helper cells might take over TFH cells functions {Schmitt, 2013; Odegard, 2008}.”
Fig 1 shows that the baseline of H3N2 in ustekinumab group is much lower than the control group. What causes the difference? Due to the frequency of vaccination or this may reflect a reduced memory response in ustekinumab group?
Response of the authors:
It is true that there was a significant lower baseline GMT in the ustekinumab group compared to the healthy controls which were healthcare workers. This difference is mainly due to a high baseline titre in the healthy control group. Table 1 shows there was no significant difference in vaccination history between the groups. Another possible explanation might be that healthcare workers were more exposed influenza than others. Since different baseline titres between study groups are a common issue in clinical vaccination studies, we aimed to correct for the high pre-vaccination titre by using a correction formula introduced by Beyer et al. described in the method section (lines 145-148). Results are shown in the results section (lines 232-236) and Table 4 (post-correction GMT). Although a reduced memory response might also be an explanation, there was no significant difference in baseline titres in the other two strains.
This limitation was added to the discussion section (lines 363-367): “There was a significant higher baseline GMT for the A/H3N2 strain in health care workers compared to the ustekinumab group. Since the influenza vaccination history was comparable between the three study groups, this might be explained by a higher exposure to influenza in health care workers {Kuster 2011}. By using a lineair regression formula we were able to correct for this possible confounder.
With the understanding that IL-12/IL-23 play a role in regulating Tfh responses, it is critical to examine the isotypes of antibody response in all treatment group. While the total HI may not be affected by ustekinumab treatment, the antibody composition in different treatment groups may be very different.
Response of the authors:
We agree with the reviewer that we cannot conclude anything about the isotypes of the antibody responses and added this statement to the limitations (line 370-372) saying:“The HI assays showed that functional antibodies are present in ustekinumab-treated patients, but we cannot conclude anything about the isotypes of the antibody response.”)
However, the HI assay is a functional assay, meaning that the binding function of present antibodies is tested by adding serum to antigen preparations. If no agglutinations happens, the antigens showed high binding-capacities. So the results does imply that the antibodies are functional and protect against the separate strains of the influenza virus in the vaccine the subjects were vaccinated with.
The authors also examined cellular responses by examining cell proliferation (Fig 3), which provided very limited information. Compared to the baseline, even healthy control group did not show enhanced proliferation, which is concerning. Intracellular cytokine staining is a preferred method to examine effector functions of CD4 and CD8 T cell responses.
Response of the authors:
We agree with the reviewer that ICS would have been useful. However, we only had a limited amount of PBMCs, so this will be difficult to redo. Therefore, we now included this in our limitations in the discussion section (line 372-375), saying: “With the T-cell proliferation we showed comparable proliferation of influenza-specific CD4+ and CD8+ T cells in all study populations, but the effector functions of CD4+ and CD8+ T cells are still unknown. Therefore, the performance of an intracellular cytokine staining would have been of additional value.”
All figures should include statistical analysis with proper labels.
Response of the authors:
We thank the reviewer for pointing out that the figure legends need some clarification. In Figure 1 and Figure 3 the HI assays and stimulation indexes are depicted for each individual in the study population without using statistics. We changed text and added the abbreviations for the time points. Statistical analyses of the difference between GMT in the three study groups are shown in Table 4 and Figure 2 and its legends explain the type of analysis used. The results of statistical analyses for differences between the study groups for the stimulation indexes are written in the results section (lines 253-257). We used Kruskal Wallis tests for comparing the three groups and Mann-Whitney-U tests for comparing two groups as described in the method section (lines 157-160). We hope that tables and figures are better clarified with the current adjustments.
Reviewer 2 Report
The main conclusion from this work is that anti-IL12/23 treatment (versus anti-TNF-alpha) of inflammatory bowel diseases does not negate influenza vaccine efficacy. This conclusion is based on the outcome adequately shown on the basis of several analytical parameters, i.e. HI, antibody titer and T-cell proliferation assays. The manuscript is well-written scientifically and language-wise. The conclusion drawn is weakened by the small sample size at authors' own admission. The work may be of clinical relevance, but the focus is not on the vaccine itself. It is the decision of the editorial office as to whether this type of work is suitable or acceptable for publication in the journal of "Vaccines".
Reviewer 3 Report
Doornekamp et al. describe that CD patients undergoing anti-IL12/23 treatment respond to influenza vaccination similarly to healthy individuals.
The paper is very well-written and the data are presented clearly.
In my opinion, the paper is an excellent candidate for its publication in "Vaccines" even without further corrections.
Reviewer 4 Report
The study by Doornekamp et al entitled "High immunogenicity to Influenza Vaccination in Crohn`s Disease Patients treated with Ustekinumab" is a very important study about the effectiveness of vaccination for patients with IBD while being on immunomodulator therapy.
The paper is well written and the data are presented accordingly and clearly structured.
As the authors state themselves, the study has some limitations such as low number of patients included or the correlation of the data with longterm clinical outcomes (morbidity due to influenza).
However, the data are of major clinical impact in every day routine. Therefore, I recommend to accept the paper in its present form.
Author Response
The study by Doornekamp et al entitled "High immunogenicity to Influenza Vaccination in Crohn`s Disease Patients treated with Ustekinumab" is a very important study about the effectiveness of vaccination for patients with IBD while being on immunomodulator therapy.
The paper is well written and the data are presented accordingly and clearly structured.
As the authors state themselves, the study has some limitations such as low number of patients included or the correlation of the data with longterm clinical outcomes (morbidity due to influenza).
However, the data are of major clinical impact in every day routine. Therefore, I recommend to accept the paper in its present form.
Response of the authors:
We thank reviewer #2 for the positive feedback and the underlining of the importance of our work. We appreciate his or her time spent on reviewing our manuscript.
Reviewer 5 Report
The authors have performed a prospective cohort study comparing the immune response to the seasonal influenza vaccine amongst patients with Crohn’s disease treated with ustekinumab, adalimumab and healthy controls (health care workers). No difference in seroconversion rates were noted between healthy controls and ustekinumab and actually. While the numbers in the study were small the findings are important and reassuring for clinicians as there is great uncertainty in this area given the limited data. I have a few queries.
Was a specific location for intramuscular insertion of the vaccine specified?
Are details of the disease phenotype (non-stricturing, non-penetrating or structuring or penetrating disease) known for the cohort? This should be reported in Table 1.
Are the % of patients on escalated doses of adalimumab and ustekinumab known? What dose of ustekinumab was routinely given?
Minor comments:
Page 1, line 41: consider changing “at risk for more severe illness due to infectious diseases” to “at risk of more severe complications of infections”
Page 2, line 70: the last word “and” should be changed to “who”
Page 2, line 72: replace “after” with “following”
Page 2, line 73: consider changing “were selected in the biobank” to “were selected from the biobank”
Author Response
Reviewer #3:
The authors have performed a prospective cohort study comparing the immune response to the seasonal influenza vaccine amongst patients with Crohn’s disease treated with ustekinumab, adalimumab and healthy controls (health care workers). No difference in seroconversion rates were noted between healthy controls and ustekinumab and actually. While the numbers in the study were small the findings are important and reassuring for clinicians as there is great uncertainty in this area given the limited data. I have a few queries.
Response of the authors:
We would like to thank reviewer #3 for a thorough review of our manuscript and the valuable comments provided.
Was a specific location for intramuscular insertion of the vaccine specified?
Response of the authors:
All intramuscular vaccines were given in the deltoid muscle of the upper arm. We now specified this injection location in the manuscript (line 88-89).
Are details of the disease phenotype (non-stricturing, non-penetrating or structuring or penetrating disease) known for the cohort? This should be reported in Table 1.
Response of the authors:
We agree with the reviewer that data on disease location and disease behaviour would be a valuable addition. We collected this data using the Montreal classification. These data were added (See Table 1). There was no significant difference in disease location and disease behaviour between the two study groups. We added the used classification system in the method section (lines 80-81) and the results of the disease location in Table 1.
Are the % of patients on escalated doses of adalimumab and ustekinumab known? What dose of ustekinumab was routinely given?
Response of the authors:
We agree with the reviewer that more detailed information on medication dose is insightful. Medication dose was heterogeneous in both the adalimumab and ustekinumab group, making it difficult to compare and draw other conclusions. Ustekinumab dose was routinely given once in 8 or 12 weeks and Adalimumab once in 2 weeks. Medication dose was determined at the discretion of the treating physician, however in both groups a minority of patients were escalated or de-escalated. We added medication dose in both study groups in the methods section (lines 81-84): “We collected medication use including dose at moment of vaccination. Ustekinumab was routinely injected in a dose of 90 mg every eight weeks or twelve weeks and adalimumab in a dose of 40 mg once every two weeks, defined as standard dose. More frequent injections were classified as escalated dose.” Data on escalated doses were added in Table 1 and in the results section. (lines 182-184): “In the ustekinumab group 1 patient was injected every seven weeks and 1 patient every six weeks. In the adalimumab group two patients were injected weekly, two patients every ten days and one every four weeks.” In the discussion section (lines 359-361) we added a comment on medication dose: “The heterogeneity in the dose of medication in both adalimumab and ustekinumab users at moment of vaccination might have influenced the results.”.
Page 1, line 41: consider changing “at risk for more severe illness due to infectious diseases” to “at risk of more severe complications of infections”
Page 2, line 73-74: the last word “and” should be changed to “who”
Page 2, line 75: replace “after” with “following”
Page 2, line 76: consider changing “were selected in the biobank” to “were selected from the biobank”
Response of the authors:
We changed these sentences accordingly.
Round 2
Reviewer 1 Report
The authors have adequately addressed my comments and concerns.